# Potential Immunologic and Integrative Methods to Enhance Vaccine Safety

**DOI:** 10.3390/vaccines10071108

**Published:** 2022-07-11

**Authors:** Alan M. Dattner

**Affiliations:** Integrative Dermatology and Medicine, Sarasota, FL 34231, USA; doc@holisticdermatology.com

**Keywords:** vaccine safety, COVID-19, long COVID, cross-reactivity, leaky gut, screening, integrative medicine, lower dosage

## Abstract

Vaccine safety is measured by the disease protection it confers compared to the harm it may cause; both factors and their relative numbers have been the subject of disagreement. Cross-reactive attack of analogous self-antigens modified by dietary and microbiome factors is one of the poorly explored likely causes of harm. Screening for that and other risk factors might point out those most likely to develop severe vaccine reactions. Cooperation from those with opinions for and against vaccination in data gathering and vetting will lead to greater safety. Screening should include an integrative medical perspective regarding diet, microbiome, leaky gut, and other antigen sources. It might include emerging electronic technology or integrative energetic techniques vetted ultimately by cross-reactive lymphocyte testing or genetic evaluation. The knowledge gained from evaluating those with reactions could enhance the screening process and, since similar antigenic stimuli and reactions are involved, help long COVID sufferers. Centers for early identification and rescue from vaccine reactions could lower morbidity and mortality, and increase the percentage of people choosing to be vaccinated. Additional platforms for boosting; using lower dosage; other routes of administration, such as intranasal or intradermal needles; and possibly different antigens could make it easier to vaccinate globally to address the new variants of viruses rapidly arising.

## 1. Introduction

I have no doubt that vaccines have protected me and the global population from endemic and pandemic diseases, with greater reduction in morbidity and mortality than the harm they produced. I also believe that there are increasingly widening gaps in protection and risks in vaccination that require enhancement of the entire process of protecting against current and upcoming infectious agents. Cooperative evaluation of immune, environmental, integrative, and other individual specific variables in severe reactors would help develop screening to predict future at-risk individuals. Better screening and intervention method availability, including evaluation of exposure to inflammatory gut antigens and flora, could reduce long-term vaccine sequelae. Based on the rapid progress made so far, one could surmise that other vaccine antigens, adjuvants, lower dosages, and routes of sensitization, could potentially increase vaccine safety, protect against new viral variants, be more acceptable to the public, and more easily allow global protection.

## 2. Controversy

There is a heated controversy regarding the use of vaccines to protect against endemic and pandemic infections, including against COVID-19 [1], that has generated exaggerated and questionable claims on both sides of the issue. This has disrupted combined progress toward what I see as a safer and more optimal comprehensive method of disease control with vaccines.

## 3. Vaccine Efficacy

Vaccination has helped control or nearly eliminate widespread infectious diseases such as smallpox and more recently cholera, typhoid fever, plague, diphtheria, tetanus, and polio and has been demonstrated in studies to be safe and effective for the COVID-19 mRNA vaccines [2]. Other reviews have confirmed short-term efficacy and safety of the COVID-19 mRNA vaccines but admitted that long term effectiveness and safety is not known [3]. The centrist medical position is that sufficiently widespread vaccination would have reduced the number of deaths by over 300,000 at 100% vaccination [4]. However, by the very nature of the process, there have been a small minority who have experienced illness either soon or late after—including neurologic injury [5] or myocarditis—and have suspected the vaccination as the cause. Those concerned about vaccine injury have pointed out sequential relationships and other data, putting suspicions on vaccination.

The new mRNA anti-COVID vaccines have slowed the spread of early variants and greatly reduced severe illness requiring hospitalization, reducing overwhelming pressure on hospitals, and initially have allowed a return to work and social interaction. This protection has not seemed to be as effective for preventing acquisition or transmission of rapidly arising new strains, or lasting as long as the protection from initial vaccines [6]. Furthermore, the side effects and deaths that, although mostly unexamined and unproven from a medical perspective, appear to have sequentially followed vaccination, have alienated people, compounded by the population polarization that has made it harder to find long-term solutions to the pandemic problem. While acknowledging the efficacy and speed of development and deployment of the mRNA vaccines [7], we need to greatly improve on prevention and treatment of vaccine reactions, and to develop a new, safer, and more acceptable platform for vaccination and prevention that is effective for the long run and for preventing infection by rapidly changing viral variants. This is crucial for protecting humans and for allowing a return to a more normal social and industrial society.

An article discussing evidence from “WHO’s Global Advisory Committee for Vaccine Safety during 1999–2019”, concludes that none of the key claimed safety related issues, including thiomersal and aluminum adjuvants, and auto-immune conditions have evidence of being related to vaccines [8]. Nevertheless, the experience of individuals or families dealing with severe or lasting reactions after vaccination, and the very nature of the sensitization process, leaves them doubting the science. Other studies have shown that 80% of some disorders ascribed to vaccination follow a familial pattern, suggesting that a genetic link explains their occurrence [9]. Yet we have not identified or screened for this component, despite the fact that most of those articles state that further exploration of interaction between genetic and environmental links is necessary. I write this while acknowledging the dilemma I struggle with in discussing potential side effects of vaccination which I believe to be personally and globally worth the risk.

The gap between the perspective of the vaccine oversight organizations and those who oppose vaccination has become so extreme that important data and investigation of adverse reaction susceptibility data gathering and evaluation is not being done. Favorable studies reassure the population about vaccination safety, but investigation of the CDC’s Vaccine Adverse Event System (VAERS) reports is incomplete enough to invite questioning. VAERS is a site in which anyone can report a presumptive adverse reaction to a vaccine. This site is not monitored or evaluated for veracity. Results are obtained by selecting a number of variables to yield individual and collective data. For example, at one time, the CDC stated that only 3 of the 6000–12,000 deaths attributed to COVID vaccination on VAERS have been proven to be related. More recently, there were 14,680 reports of death from COVID vaccines out of 581 million doses administered. The interpretation of VAERS data from the anti-vax side is that there have been 1,277,980 injuries from COVID Vaccine and that the 28,312 or so VAERS-listed vaccine deaths [10] have been grossly underestimated. There is little room for rational discussion of such differences. On a more relevant note, glancing at what one dizziness reactor ate before and after his vaccination, reveals a high sugar allergenic diet that would have made me sick without any vaccine. It is time that the efforts from both sides move to further verification and determining of the parameters leading to severe sequelae or death from an immunologic and integrative perspective, so that we better know who, when, and how an individual can safely be vaccinated. It is time to make billion dollar increases in the efforts and resources available for early identifying and reversing both long COVID and vaccine reactions. Additionally, we need to begin to modify or add to the vaccines that afforded such early success to more effective additional ones with safer dosages, routes of administration, and flexible control of new variants. 

We also need to explore other existing modalities, which combined, and in the right circumstances, will enhance the effectiveness of vaccines in preventing infection, virus transmission, or treatment. The value of other drugs and supplements to protect against or treat COVID is anecdotal, and has been vigorously denied by the FDA and medical establishment [11], rather than explored as to whom and when and in what combinations they might prove valuable to enhance protection and reduce transmission, in conjunction with vaccination. 

Not all drugs and supplements are effective, and some—such as Ivermectin—may have been more effective against earlier COVID variants than newer ones.

Although not all available substances may be effective alone or in combination with most people, it is time to include a multi-dimensional evaluation that includes a gut dysbiosis as well as relational immune status evaluation in determining the likelihood that various diets, supplements, and drugs—alone or in combination—are effective at preventing or treating COVID and vaccine reactions, and long COVID. Vetting simple tools such as clinical evidence of gut dysbiosis and associated symptoms in past and ongoing reactors, and anecdotal reports of effective diets, remedies, and the characteristics of those helped and not would create a helpful picture of where these combinations are useful. 

Governmental health agencies and funders appear to allocate as if the fear of commonly available treatments driving people away from vaccines is greater than the need for discerning in what circumstances and individuals they will be effective. Anti-vaxers wave the flag of every dramatic reaction or death following COVID vaccination rather than exploring the dysbiosis, integrative, and other clinical aspects of the unfortunate person in order to create an algorithm for who is at greatest risk. The cost and effort in the direction of exploring circumstances and simple preventatives and treatments will be more than compensated by the boost it will give to our economy [11], and the livability of our planet. Lack of cooperation is no longer a viable option. It does not take more than a trip to the store, or the avoidance of one, to understand the ongoing economic devastation due to the COVID-19 pandemic.

Getting a majority of our planet safely protected from each new COVID or other viral variant that makes human contact risky, and preventing long COVID, is crucial. Indeed, the better collection of data regarding those with cross-reactive attacks leading to long COVID will benefit those who may have vaccine reactions due to components of the vaccine such as spike protein, which can stimulate tissue-specific attacks in those with organ-specific molecules that—along with their tissue type—mimic the stimulators and initiate attack. Safer vaccines will encourage more people to get protected.

## 4. Cross-Reactive Attack

I believe that one of the major causes of post-vaccine reactions is cross-reactive TH1 lymphocyte attack by immunologically vaccine-similar molecular targets on the host. This is related to the very mechanism by which the vaccine induces immunity to the bacteria or the virus it is protecting against [12,13,14]. Once T-cells have been immunized against a specific batch of molecular targets from a viral infection or vaccine, they are free to attack similar targets they find throughout the body. My research with Levis and other subsequent research suggests that these T-cells initiate attack and inflammation against a very specific target or “epitope” which bears characteristics of the HLA [15], tissue specificity, and the viral signature involved. COVID vaccination has been followed by autoimmune reactions in various organs and specific tissues within those organs [16], including myocarditis in the heart [17,18], hepatitis-like syndrome in the liver [19], autoimmune bullous disease in the skin, thrombosis of platelets [20], and neurologic disorders [21].

This epitope may well be initially presented to circulating T-cells by endothelial cells lining the blood vessels with the specificity of those organs [22], such as the brain or heart. It is likely that this heterogeneity accounts in part for COVID, post-COVID, and even COVID vaccine reactions attacking different organs in different people [23,24]. Severity of COVID illness has been linked to intestinal dysbiosis [25]. The presenting structure on these cells may be further modified by seeding the vascular and lymphatic system with circulating chemical, food-derived, or microbiome-derived molecules such as LPS from the 30 trillion colonic microorganisms. Microbial chemical epitope modulation would increase the diversity of the presenting epitopes on organ specific vascular endothelial cells [26,27] to greatly increase the chance of cross-reactive stimulation and destructive attack leading to COVID symptoms, vaccine reactions, or to long COVID [28]. Increased likelihood of adverse vaccine reactions, as with other autoimmune reactions, could in part be due to these molecules translocated into the circulation or lymphatics through a “leaky gut” [29].

Besides genetic mechanisms for modifying epitopes, non-genetically post-translational epitope modification expands epitope diversity and has been associated with autoimmune diabetes [30]. That modification has been observed due to citrullination [31], from *Porphyromonas gingivalis*, and correlates with Rheumatoid antigen. Lipopolysaccharides (LPS) more often are derived from the gut microbiome than that of the oral cavity, and have been observed to increase inflammatory activity [32]. It is my hypothesis that yet unidentified enzymatic, antigenic, or other post-translational effects caused by some of the estimated million different gut microorganisms could also be enhancing immune responses or modifying tissue-specific epitopes to increase diversity and immune attack by cross-reactive lymphocytes stimulated by COVID vaccine, or some molecular aspect of the COVID disease process.

Presenter cell modification in the clinical hypothesis above explains why there is a conflict with the conclusions of an in vitro study, showing that the spike protein does not share tissue homology with myocarditis-associated antigens [33]. Numerous studies have demonstrated a slight increase in myocardial reactions and myocarditis, post vaccination—especially in young men [34,35]—and antigen modification of cardiac endothelial presenter cells [36] would invalidate that conclusion, and furthermore suggest that such modification is key to the damage done by SARS-CoV-2 and its antigens. The latter article argues that the protective benefit of the vaccine is greater than the slight risk of damage.

Reduced severe and long-lasting vaccine reactions would seem to reduce the population and vehemence with which vaccines are rejected. Although acute allergic reactions are well managed by our current system, there is no network of sophisticated centers to aid in screening before vaccination, nor in analysis and treatment of long-term post-vaccination reactions or preventing post-vaccination death [37].

## 5. Value of Greatly Improved Pre-Vaccine Evaluation

Better screening is needed to identify those who should not have the vaccine without more sophisticated testing—including dose, route, or content modification—and those who should be carefully watched sequentially after vaccination. Much of this information, as listed above, as well as questions on immune hyperactivity and dysregulation, could be gathered from a questionnaire based on a model from integrative medicine, now beginning to appear in mainstream literature, focused on diet, probiotic benefits, Candida overgrowth, identification, and treatment of intestinal dysbiosis and leaky gut [38]. The model I propose for assessing inflammation-related additional antigen exposure—which has been used in my practice for over 30 years to deal with patients with inflammatory and autoimmune disorders [39,40]—would lead to a number of successive questions that could be first administered by a computer and then by a savvy nurse or technician. 

Risk of vaccine reactions, determined by such questioning, could funnel people toward emerging electrodiagnostic [41,42] and energetic techniques followed by lab tests such as antibody levels, cytokine profiles, genetic profiles [43], and epigenetic assessment chips. Cross-reactive testing of lymphocytes stimulated by vaccine components and the resultant cells tested against individual allogenic or autologous cells from various organs would be one of the gold standards, along with genetic profiling and in silico comparison, for assessing whether the vaccine was likely to damage specific organs. It would be more accurate if carried out with circulating LPS, microbes, and other circulating epitope modifiers. Epitopes are immunologically recognized accessible molecular structures which, by their chemical nature, can be altered or modified by chemical moieties contacting them in the circulation. Modification of epitopes by bacterial products by processes including citrullination [44] and other signals from the microenvironment [45].

An effective screen and treatment system would reassure the questioning public that the vaccine system ‘has their back’ and will support them against adverse vaccine reactions. This evaluation might verify phenotypic features of such reactions that are far cheaper and easier signals of need for immediate care. For example, evidence has emerged that a headache one week after COVID-19 vaccination may be a sign that cerebral venous thrombosis has been induced by the vaccine [46]. There also needs to be a weighted algorithm for testing and choosing appropriate medical and integrative remedies for terminating such reactions to protect the individual.

Anecdotal reports supportive of this hypothesis include finding increased cerebrovascular sequelae in older Veterans with IBD or IBS, both of which conditions would increase the intestinal bacterial microbiome entering the circulation and lymphatics, to increase diversity of presented epitopes to be attacked by T-cells stimulated by the SARS-CoV-2 infection. COVID disease severity and ongoing symptoms may be related to disturbed microbiome in the gut [47]. Gut microbiome modulation may moderate this effect [48]. A comprehensive review suggests that microbiome alterations could diminish the number and severity of myocardial disorders associated with COVID infection [49].

## 6. Immune Regulation Disturbances

Another risk is activating and/or dysregulating the immune system, which interferes with both recognition choices and inhibition of organ destructive immune processes by, for example, T-Reg cell activation and corticosteroid release. Stress, illness, and viral infections are immune dysregulators. SARS-CoV-2 virus is a dysregulater which strongly activates intracellular immune factors, such as NFkappaB [50], to prompt the cell to produce more virus containing vesicles. Viral particles are then released on destruction of the cell and spread to adjacent cells and through the body. Virus protective interferon production is inhibited by the virus [51]. When the acute viral process has calmed down, the virus has infected or altered surface components of cells of various organs of the body, for potential recognition. Circulating lymphocytes have become hypervigilant, specifically sensitized and ready for aggressive attack against these and analogous recognized targets. This can lead to specific organ attack and cytokine storm. It is analogous to pressing down the brakes and gas pedal, revving the engine, and then taking your foot off the brakes. The car takes off risking skidding out of control.

Treatment of allergic vaccine reactions is well known. Potentially serious reaction treatment may consist of drugs including biologics, depending on severity and rapidity of symptom evolution, and evaluation in use. In lesser acute situations, they also may consist of herbs, supplements, and diet that support T-reg formation and amelioration of Small Intestinal Bacterial Overgrowth (SIBO) and leaky gut. SIBO refers to the overgrowth or backflow of colonic organisms into the small intestines, which if hyperpermeable—as in leaky gut—allows LPS, bacteria, yeast, fungus, viruses, and other eventual components of the stool to leak into the lymphatics and circulation and be distributed throughout the body. This greatly favors increased organ target antigen presentation diversity, and multiple toxic, immunologic, metabolic, epigenetic disruptions. Correcting those issues by cutting down on high glycemic foods to reduce Candida, treating other sources of dysbiosis, and resolving SIBO can help. Anti-inflammatories such as quercetin and baicailin from *Scutelaria baicalensis* and herbs that open up microvasculature are also helpful.

Is all of this expensive? Yes, but not in comparison to the billions that have been thrown around, the cost of care for those developing severe COVID who have avoided vaccination, and those who have developed long COVID because of a case of mild or severe COVID that could have been prevented. The greatest value of all of this evaluation, data gathering, assessment, and prevention of vaccine reactions is that it is potentially applicable to identifying, preventing, and treating long COVID, which in my view and the medical literature is the result of a cross-reactive-specific and general immune attack, as described above. Long COVID is estimated to affect about 10–30% of those who contract COVID [52]. Long COVID is only partially prevented by vaccination, but prevented better than with no vaccination at all. Additional measures are recommended to prevent long COVID [53]. This takes a big bite out of the labor force, and the people in this country and all over the Earth who are necessary for supply chains, manufacturing, and farming to deliver products that people depend on. That cost is reflected in both inflation and inability to obtain needed products. Even with the absence of liability for vaccines, we all share in the cost of the consequences, and through governmental funding need to generously devote the billions necessary to avoid those costs.

## 7. Potential Low-Vaccine-Dose Efficacy

When I was a visiting scientist at the NIH studying lymphocyte reactivity to microbial antigens, we had some fascinating, unexpected results. Chemical antigens such as DNCB that caused allergic contact dermatitis from sensitized individuals caused lymphocyte stimulation and growth, and those resultant stimulated lymphocytes reacted and grew even more strongly when stimulated by cells treated with DNCB. However, when we stimulated lymphocytes with the optimal concentration of microbial antigens such as tetanus toxoid, the resultant lymphocyte population did not react with growth until we decreased the stimulating dose to 1/250th or 1/1000th of the optimal level [54]. Clinically, quarter doses of the Moderna vaccine were found to give similar immune protection data including “Vaccine-specific CD4^+^ T cell, CD8^+^ T cell, binding antibody, and neutralizing antibody to COVID infection, and enhancement of Coronovirus cross-reactive CD4^+^ T cells” [55]. Following these publications, the Moderna booster dosage was lowered to half the original dosage.

It is possible that in some individuals or in repeated sensitization to new and emerging strains, a 10th, 100th or even 1000th, of viral antigen signatures could be sufficient for vaccination. That would allow cheaper, easier production of sufficient amounts of vaccine and far safer vaccination dosages to protect everyone on the planet. Other routes of sensitization could be used that mimicked natural immunity buildup, such as intranasal administration of antigen [56,57]. Furthermore, other routes of sensitization using microneedles could be considered [58], depositing the antigen intradermally to enhance TH1 immunity.

I commend the companies that initially rapidly produced an effective vaccine which reduced the early waves of infection, severe illness, and mortality from COVID. They were designed to give the quickest protection to the initial viral variant. In this era of chronic and changing variants of COVID, influenza, and other viruses, modified, more easily produced, lower dose, and differently administered vaccines should be explored with the same urgency and funding as was used to produce this initial preventative vaccine.

Classically, sensitization to produce T-cell immunity was carried out by intradermal injection of antigens combined with adjuvant into animals. Microneedles would better mimic this intradermal route. T-cell sensitization and immunity is an important component of sensitization, because the T-cells regulate antibody immunity and have better plasticity of recognition of virus than antibodies, and many subsets have a role in viral immunity—including follicular T helper cells, memory T cell, regulatory T cells, and the interaction of subsets [59].

My only wish is for the greater health and greater good of mankind, in the face of viral and other health threats that have come upon us and are threatening our health, social interactions, and economic wellbeing as a population. I am a long-time classical immunologist, and former Visiting Scientist at the NIH who became an integrative physician because of my insights gained from researching and understanding the complex nature of the individuality of immune recognitions by human lymphocytes. I have been immunized against COVID and many other infectious agents, and believe that risk from vaccines for me is lower than risk from some diseases I may be exposed to. I understand and have followed the information, estimates, and explanations of both camps—the vaxers and the anti-vaxers—and believe that there is far more information to gather and evaluate, and there are some valid points to consider, and distortions of information from both groups. I favor vaccination personally—despite the risks—and professionally, with the safeguards I have suggested.

## 8. Conclusions and Future Directions

I believe from history, my experience, and the medical literature, that vaccines provide more benefit than their risk, and that risks do exist. Those risks could be decreased by progressive screening before vaccination and after, and with appropriate rescue techniques.

Effective screening evaluation would be greatly enhanced by cooperative data gathering from those with reactions. Questions would combine an immunologic perspective with an integrative medical model covering gut microbiome, leakage, and disturbance as well as other recognized parameters. Pre-vaccination dietary and environmental recommendations would likely emerge to reduce the chances of vaccine reactions and severe COVID. Better screening and post vaccination follow-up could reduce reactions and vaccine hesitancy, increasing the percentage of the population who get vaccinated.

Rapid emergence of resistant COVID strains suggests exploring lower vaccine doses and a more flexible, cheaper, safer, less invasive platform for global vaccination, following the initial success with the mRNA vaccines. If this pandemic is to be controlled, it is time for major funding to be allocated and cooperative data collection to begin towards the medical, integrative, and immunological approaches listed above.

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
