# Peer review of "Potential Immunologic and Integrative Methods to Enhance Vaccine Safety"

_vaccines, 2022, doi:10.3390/vaccines10071108_

Round 1
Reviewer 1 Report
The current perspective manuscript from Alan Michael Dattner provides interesting insight into his angle on how to implement, establish and utilize future pre-vaccination screens in order to identify potential risk factors or modalities (like leaky gut syndrome or genetic predispositions) that could predict adverse vaccination reactions. Alternative vaccination regimes like intradermal microneedle patches as well as intranasal vaccination are briefly discussed to complement current vaccination strategies allowing the use of less antigen material reducing adverse vaccine reactions. Furthermore, intranasal vaccines would improve the local tissue-based immunity in the upper airways thereby reducing the circulation of COVID-19 and similar diseases.
The manuscript covers a very relevant and intriguing topic with some highly valuable suggestions for future directions of research and vaccination management. However, there are currently some inconsistencies and flaws in the paper that need to be addressed prior to publication:
Major points
1) One of the major weaknesses of the manuscript is the current use of references. While the vast majority of the cited works are mostly current and the number of references is also very appropriate, they actually do not sufficiently support the claims or statements made in several cases. For example in the section on cross-reactive attack (line 126-129) you claim that cross-reactivity might be to blame for many adverse vaccination reactions. While this sounds plausible, the cited work only refers to cross-reactivity between different pathogens and does not thematize anti-self or auto-immune reactions. Please find matching references for this.
2) Line 137-138: Please include a reference here to demonstrate that different patients experience reactions in different organs.
3) Line 138 and related to 1): please give a reference for the link between severity of COVID illness and intestinal dysbiosis.
4) Line 142-145 and related to 1): please also find a more relevant reference to support the core message of the tissue heterogeneity potentially causing anti-self immune reactions.
5) Line 145-146: Is there any evidence of leaky gut or other gastrointestinal issues correlating with adverse vaccine reactions? I would find this theory very plausible. Please refer to a reference if this is the case.
6) Line 159: Please find a more relevant reference than this historic perspective of the Indian vaccination infrastructure and management system mostly covering the 19th and 20th century.
7) Line 174: You are citing Minami et al. here again on Organ/Tissue-Specific Vascular Endothelial Cell Heterogeneity . Please replace with a better fitting reference.
8) Line 180: Please describe in more detail what you mean with epitope modifier. To my knowledge epitopes can be modified when posttranslational modifications are added, or antigen sequence changes occur. I do not see this happening from circulating LPS or microbes. I'd rather believe that these epitopes from LPS or microbes would be picked up by APCs and other antigen presenting cells and be presented to B and T cells potentially leading to immune reactions against the microbiome.
9) Line 194-196: The used reference is from 2018 and does not mention COVID. Please correct to a reference supporting this sentence.
Minor points
10) In the abstract it is not very clear to me how you jump from vaccine safety to long COVID. While this gets clearer in the main text, as you elaborate a bit more on your hypothesis that long COVID arises from cross-reactivity or auto-immunity. Please refer to this thesis already in the abstract, so people can follow your line of argumentation better.
11) I suggest to refer to PMID: 25937189 as a helpful review on adverse vaccination reactions, vaccine safety and vaccine design.
12) Line 85: Please write out in full what VAERS stands for and briefly explain for the nonUS-based readership.
13) Line 87: Is it 6-12,000 or 6,000-12,000 deaths? Please use the latter to make things clearer.
14) Line 91-92: This link did not work for me. Please replace with a functional link.
15) Line 99: I believe there are considerable resources already being invested in countering long COVID, so rephrasing to indicate a more intense effort is needed in this area might be more appropriate.
16) Line 112: I suggest here to at least briefly mention that it is clear that not every commonly available and licensed therapeutic can and should be evaluated as a potential COVID treatment, but promising alternatives that have shown benefits in anecdotal cases or mouse studies should be explored further and efforts in testing these ramped up.
17) Line 115-116: This is purely speculative unless you provide a number of references supporting your statement. It could also be that spreading resources over too many research fields at the same time could lead to very limited outcome in the end. Please rephrase, provide evidence or clearly indicate this as a personal opinion.
18) Line 216: Please briefly explain what SIBO stands for.
19) Reference section - ref 4: The used references is a preprint and should be marked as such.
20) Reference section – The paper of Minami T et al. is listed three times in the reference section. Please correct this, so it is listed only once.
Reviewer 2 Report
The sectioning of the paper seems a bit wired. There is only one main section named ‘Introduction’ and seven sub-sections.
The references are not properly numbered. For instance, between ref.18 and ref.19 there is one un-numbered paper.
The formatting of reference is also non-consistent throughout the manuscript. For example, ref. 36 starts with title and then author names, while ref. 37 starts with author names and then the other information.
The written style of the perspective is more like an essay rather than a scientific paper published in international journals. I am not sure whether this is common in the journal Vaccines.
The abbreviation GACVS is defined but not mentioned later. In this case, it should not be defined at all. By contrast, VAERS is mentioned many times but not defined.
Round 2
Reviewer 1 Report
The author hast addressed the vast majority of my recommendations or comments. I therefore highly recommend publication of the article.